# Effect of Training-Detraining Phases of Multicomponent Exercises and BCAA Supplementation on Inflammatory Markers and Albumin Levels in Frail Older Persons

**DOI:** 10.3390/nu13041106

**Published:** 2021-03-28

**Authors:** Adriana Caldo-Silva, Guilherme Eustáquio Furtado, Matheus Uba Chupel, André L. L. Bachi, Marcelo P. de Barros, Rafael Neves, Emanuele Marzetti, Alain Massart, Ana Maria Teixeira

**Affiliations:** 1University of Coimbra, Faculty of Sports Sciences and Physical Education—(FCDEF-UC), 3040-248 Coimbra, Portugal; rsneves.prof@gmail.com (R.N.); alainmassart@fcdef.uc.pt (A.M.); ateixeira@fcdef.uc.pt (A.M.T.); 2Research Centre for Sport and Physical Activity, CIDAF-FCDEF-UC, 3040-248 Coimbra, Portugal; matheusuba@hotmail.com; 3Health Sciences Research Unit: Nursing (UICISAE), Nursing School of Coimbra (ESEnfC), 3000-232 Coimbra, Portugal; 4Department of Otorhinolaryngology, ENT Lab, Federal University of São Paulo (UNIFESP), São Paulo 04025-002, Brazil; allbachi77@gmail.com; 5Post-Graduation Program in Health Sciences, Santo Amaro University (UNISA), São Paulo 04829-300, Brazil; 6Institute of Physical Activity Sciences and Sports (ICAFE), Interdisciplinary Program in Health Sciences, Cruzeiro do Sul University, São Paulo 01506-000, Brazil; marcelo.barros@cruzeirodosul.edu.br; 7Fondazione Policlinico Universitario “Agostino Gemelli” IRCCS, 00168 Rome, Italy; emanuele.marzetti@policlinicogemelli.it; 8Università Cattolica del Sacro Cuore, 00168 Rome, Italy

**Keywords:** inflammaging, cognitive impairment, cytokines, protein intake, physical frailty

## Abstract

Nowadays, it is accepted that the regular practice of exercise and branched-chain amino acids supplementation (BCAAs) can benefit the immune responses in older persons, prevent the occurrence of physical frailty (PF), cognitive decline, and aging-related comorbidities. However, the impact of their combination (as non-pharmacological interventions) in albumin and the inflammatory markers is not fully understood. Therefore, we investigated the effect of a 40-week multifactorial intervention [MIP, multicomponent exercise (ME) associated or not with BCAAs] on plasma levels of inflammatory markers and albumin in frail older persons (≥75 years old) living at residential care homes (RCH). This study consisted of a prospective, naturalistic, controlled clinical trial with four arms of multifactorial and experimental (interventions-wahshout-interventions) design. The intervention groups were ME + BCAAs (*n* = 8), ME (*n* = 7), BCAAs (*n* = 7), and control group (*n* = 13). Lower limb muscle-strength, cognitive profile, and PF tests were concomitantly evaluated with plasma levels of albumin, anti- and pro-inflammatory cytokines [Interleukin-10 (IL-10) and Tumor Necrosis Factor-alpha (TNF-α) respectively], TNF-α/IL-10 ratio, and myeloperoxidase (MPO) activity at four different time-points: Baseline (T1), after 16 weeks of multifactorial intervention (T2), then after a subsequent 8 weeks washout period (T3) and finally, after an additional 16 weeks of multifactorial intervention (T4). Improvement of cognitive profile and muscle strength-related albumin levels, as well as reduction in the TNF-α levels were found particularly in ME plus BCAAs group. No significant variations were observed over time for TNF-α/IL-10 ratio or MPO activity. Overall, the study showed that MIP triggered slight alterations in the inflammatory and physical function of the frail older participants, which could provide independence and higher quality of life for this population.

## 1. Introduction

Aging is characterized as a natural degenerative process strongly linked to diminished immune efficiency, and also to enhanced inflammatory responses, and thus, to higher risks of infections in older persons [1]. The sedentary lifestyle, per se, is one of the most important contributors to age-related illness, whereas regular physical exercises (rPE)—based on hormesis principles—could chronically slow down the aging immune/inflammatory dysfunctions [2]. In this sense, reduction of systemic levels of interleukin-10 (IL-10), a classical anti-inflammatory cytokine, with elevation on Tumor Necrosis Factor-alpha (TNF-α) levels are associated with aging [3]. Although the participation in rPE programs does not stop the progression of aging [4], staying in moderate rPE programs can help making the aging process more rewarding, with lower incidence of premature chronic diseases [5]. In addition to the comorbidities outcomes, both aging and the sedentary behavior may speed up the loss of mobility and functional autonomy [6], reducing the quality of life [7], and also increasing the susceptibility to physical frailty (PF) and cognitive decline [8].

The age-related PF syndrome is defined by loss of muscle mass (and sarcopenia), by low physical activity levels, and often accompanied by low protein intake [9]. Cognitive decline, in turn, is characterized by confusion and progressive loss of memory and neuromotor skills [10]. However, these two outcomes reveal biological and phenotypic similarities, which is the reason leading to the current scientific interest in investigating populations affected by these disorders [11]. In this sense, rPE could also provide protection against both PF and cognitive decline in very old people [12], with most of these benefits related, at least in part, to changes that occur in the immune system [13]. Recent findings have shown that multicomponent exercise (ME) interventions, those that include different types of endurance, muscle strength, and balance exercises in the same session, appear to have a superior effect on cognitively and physically frail older persons [14,15].

Participation of older persons in rPE ameliorates not only antigen recognition, but also immune responsiveness in general, as some evidence has shown that increased levels of physical activity using exercise routines can even extend the protection provided by the influenza vaccine in older persons [16], as well as a regulation of systemic inflammatory status [17]. Apart from the modulating effects of rPE, nutritional habits also play an important role in determining immune and inflammatory efficiency, especially in older persons [2]. In fact, malnutrition in older population is a serious concern for health systems around the world, since it increases the risk of comorbidities occurrence with subsequent higher health care costs [7,8]. Indeed, nutritional supplementation with vitamins, antioxidants, and protein components (including isolated amino acids) have already demonstrated positive results against PF, cognitive impairment, sarcopenia and other age-related disorders [18]. 

Supplementation with BCAA, in the absence of branched-chain aminotransferase (BCAT) activity in the liver implies that a dietary supply of BCCAs would ensure an almost intact passage through the liver directly to the muscle tissue, which seems to be advantagous to restrain sarcopenia and frailty [19]. Supplementation with BCAA, especially in association with regular exercises, was demonstrated to improve muscle strength and cognitive functions in the older population, which are safe and low-cost strategies to circumvent the general limitations imposed by the aging process [20,21,22].

Among several pro/anti-inflammatory biomarkers used in the context of exercise and nutrition sciences [23], myeloperoxidase (MPO) stands out as a valid marker largely released by activated neutrophils, with potent pro-oxidative/pro-inflammatory actions 24 [24]. MPO activity also appears as a biomarker that was strongly associated with frailty and risk of mortality in a study conducted in a large community-dwelling frail octogenarians and nonagenarians [25]. Recently, a similar intervention demonstrated the slight reduction of serum MPO activity triggered by the combination of Taurine and ME in older persons [26]. Instead, albumin concentrations are currently used for the assessment of the nutritional status of an individual, and low albumin concentrations have been associated with increased mortality after correlation for age, body mass index (BMI), gender, and several chronic comorbidities [27]. In this sense, multifatorial interventions programs (MIP, exercise plus protein suplmentattion) that target to maintain (or even increase) albuminemia in older persons could characterize an important strategy to diminish the harmful effects of aging and its comorbidities [28].

Therefore, the aim of this work was to evaluate the effect of a 40-week MIPon plasma/serum pro- and anti-inflammatory markers of the immune system in older persons living in residential care homes (RCH). Furthermore, we hypothesized that ME plus BCAAs may have an impact on the systemic albumin levels, inflammatory variables, cognitive profile, and physical function of the participants.

## 2. Materials and Methods

### 2.1. Preliminary Procedures and Ethics

This is a prospective, naturalistic, controlled clinical trial (treatment vs care). All subjects volunteered to participate in the exercise classes or the supplementation programs. Consent forms were signed by the institution’s directors, the participants and their legal representatives before testing and intervention. This study was approved by the Ethical Committee of Faculty of Sport Sciences and Physical Education, University of Coimbra (reference number: CE/FCDEFUC/00282018), respecting the Portuguese Resolution (Art.°4th; Law no. 12/2005, 1st series) on ethics in human research and the Helsinki’s Declaration. This study was properly registered with clinicaltrials.gov register NCT04376463.

### 2.2. Participants Elegibility

Study participants were selected through a non-probabilistic trial (plus controlled sampling) living in public and private RCH. The eligible criteria for the participants in this study were, at the time of first screening: (i) Participants had to be 70 years old or more; (ii) physically frail and pre-frail; (iii) clinically stable with their drug therapy updated; (iv) being able to perform the Time Up and Go test in ≤50 s that indicate severe mobility independence [29]; (v) not participating in other structured rPE; (vi) not presenting any type of health condition or use medication that might prevent the functional self-sufficiency test performance or attention impairment (such as severe cardiopathy, hypertension, uncontrolled asthmatic bronchitis or severe musculoskeletal conditions); (vii) not presenting mental disorders or hearing/visual impairment that could prevent the evaluations and activities proposed, according to the institutional medical staff; (viii) not presenting morbid obesity (BMI ≥ 40). At the end of the recruitment process, 80 older persons entered the enrollment phase.

### 2.3. Participants Allocation

All the participants were selected through a non-probabilistic trial (plus controlled sampling) based on the geographical area of Coimbra, Portugal, living in public and private residential care homes (RCH) or frequenting day centres in the local community. From the 80 participants initially screened, 50 eligible participants were allocated in their respective intervention groups. However, for the specific reasons highlighted in Figure 1, only 35 participants (age = 83 ± 3 years-old) completed the 40 weeks multifactorial intervention, divided in the following groups: ME (*n* = 7), ME + BCAA (*n* = 8), BCAA (*n* = 7), and the no-regular exercise/no-supplementation control group (CG, *n* = 13). All the procedures were performed according to the Consolidated Standards of Reporting Trials (CONSORT) guidelines [30].

### 2.4. Experimental Design

This study is a four-phase prospective, naturalistic, controlled clinical trial with four arms of MIP experimental design (ME + BCAAs, BCAAs, ME, and CG). In the first phase, a baseline data collection (T1) was done followed by 16 weeks of MIP. The second phase consisted of a second data collection (T2) followed by an 8 week washout phase. Phase 3 consisted of a third data collection, followed by the resumption of the MIP for a period of 16 weeks. The last data collection took place after the 16 weeks of intervention (T4) (Figure 2).

## 3. Outcome Measures

All the assessments were performed in the morning, between 10 and 11:45 a.m. One session was used to apply a short test battery to measure biosocial, global health status, cognition profile, nutritional, physical, and physical frailty status. In the second consecutive day, blood samples were collected and stored at −80 °C until further analysis.

### 3.1. Physical Frailty Index

The phenotype of Fried’s physical frailty index was used [9]. Weight loss was assessed by a self-report of unintentional weight loss of 4 kg or more in the last 6 months. Self-reported exhaustion was evaluated by a negative concordance of question number 7 and 20 of the Center of Epidemiologic Studies for Depression scale [31]. Hand-grip strength was assessed in kilograms by a hand-held (HGT) dynamometer (Lafayette 78,010, Sagamore, United States). The best result of the two trials was used for scoring purposes. Participants who were unable to perform the HGT and those in the lowest 20% were categorized as positive [32]. The cutoff reference values for HGT of ≥29 kg for male and ≥17 kg for female were adopted. Slowness was measured by the “15 feet (4.6 m) walking test”. Based on the cutoff values of Fried’s study population, the times of ≥7 s for males and ≥6 s for females were adopted for positive scores of slowness. The best time of the two trials was used for the final scoring. Low physical activity (PA) levels were assessed by the International PA Questionnaire short version (IPAQ-SV) [33]. There are three levels of PA suggested for classification: Inactive, minimally active, and highly active. Participants classified as inactive had a positive score for this PF component. A positive evaluation in one or two criteria classified the participants as pre-frail, in three or more criteria as frail, and as non-frail when the subject scored none of the five PF indicators. The prevalence of PF was calculated to generate a frailty total score, as well as the presence of each of the five criteria of the Fried’s model (0 to 5 points). In this study, participants classified as frail (3 or more points) and pre-frail (2 points) were included.

### 3.2. Nutritional Assessment

Daily diet at the RCH was prescribed by a registered nutritionist and was provided for all the participants without any change or interference of the research staff. On the basis of the information provided, the diet was analyzed using specific tools (photographic quantification of portions, food table) for the Portuguese population [34,35,36]. Due to the relationship between the frailty status and severe decrease of muscle mass (or sarcopenia) which had already been demonstrated in several studies, the objective of this nutritional assessment was to characterize the protein consumption of the participants. In addition, the Mini Nutritional Assessment (MNA) questionnaire was applied [37,38]. This consists of 18 questions that present a maximum score of 30 points, and classifies the participants as malnourished (≤17 points), at risk of malnutrition (17 < MNA < 23.5 points), and as having a normal nutritional status (MNA > 23.5 points).

### 3.3. Lower Limb Muscle-Strength Test

The Five-Times-Sit-to-Stand-Test (5TSS test) was applied. This test assesses the functional strength of the lower limbs, transition movements, balance, and risk of falling. The participant is instructed to stand as quickly as possible five times, without stopping in the middle. In addition, the participant should be encouraged to keep his arms crossed over his chest. The instructor must count the time with a stopwatch and must count each position out loud so that the participant remains oriented. The test is stopped when the participant reaches the orthostatic position at the 5th repetition [29].

### 3.4. Clinical and Health Status

The Charlson comorbidity index (CCI) was calculated based on the registry of individual comorbidities combined with age and gender, to account for a final score [39]. The anthropometric assessment included body mass (kg) and stature (m). Body mass was determined using a portable scale (Seca^®^, model 770,Berlin, Germany) with a precision of 0.1 kg, whereas stature was determined using a portable stadiometer (Seca Body meter^®^, model 208, Berlin, Germany) with a precision of 0.1 cm. Body mass index (BMI) was calculated according to the formula (BMI = body mass/stature^2^). The standardized procedures described in previous studies were followed [40].

### 3.5. Cognitive Profile

The Portuguese version of the Mini Mental State Examination (MMSE) was used [41]. The MMSE is a 30-point scale instrument that evaluates five domains of cognition: Orientation, immediate recall, attention and calculation, delayed recall, and language. This scale classifies individuals by progressive cognitive skills: (0–9 points) severe cognitive impairment; (10–18 points) moderate cognitive impairment; (19–24 points) mild cognitive impairment; and (25–30 points) normal cognitive profile [42].

### 3.6. Biochemical Analysis

Non-fasting blood collection was done in the morning (between 10:00 a.m. and 11:00 a.m.). Blood samples were collected by venipuncture, after 15 min of individual rest in an isolated and quiet room, at the four time-points of the study assessment. The participants were asked to avoid alcohol and caffeine intake on the previous day of blood collection, and also to maintain their sleep habits during the previous night. After centrifugation at 3000 rpm at 4 °C during 15 min, plasma and serum samples were aliquoted into Eppendorf tubes and stored at −80 °C until used for the determination of interleukin-10 (IL-10), tumour necrosis factor alpha (TNF-α), myeloperoxidase activity (MPO), and total albumin concentrations. The ELISA (Thermo Fisher, Gloucester, UK) intra-assay coefficients of variability were 4.1% for IL-10 and 3.0% for TNF-α.

### 3.7. Full Characterization of the MIP

#### 3.7.1. Oral BCAAs

The BCAAs power mixture was composed of L-leucine (Leu), L-isoleucine (Ile), and L-valine (Val) in the proportion of 2:1:1 (MyProtein^®^, Cheshire, UK), accounting for 20 kcal per portion, comprising 5 grams (g) of supplement: 1.85 g Leu, 0.93 g Ile, and 0.93 g Val. The unflavored supplement was used as to not induce ingestion preferences for specific flavors. The BCCAs were diluted in 200 mL of water and given immediately after the exercise sessions to the participants in the ME + BCAAs and BCAAs groups [43]. The supplement dose was fixed at 0.21 g total BCAAs/kg/session, with individual portion sachets, administered in the morning, between 09:00 and 11:30 a.m. [44]. We opted to exclude maltodextrin or the carnosine-based placebo here, since the carbohydrate ingestion could mask the effort perception and cognitive indexes in our older persons volunteers, compared to the amino acid supplementation [45]. In addtion, carnosine, as well as other β-alanine derivatives, were shown to affect cognitive functions, including the perception of wellness, mood, and depression indexes [46]. Therefore, we decided to split BCAA-supplemented (ME + BCAAs and BCAAs) and BCAAs-absent groups (ME and CG) according to the proximity between the residential care homes (RCH), where the ME programs were effectively applied. No communication was reported between volunteers from the BCAA-supplemented and no-BCAA supplemented groups in our study.

#### 3.7.2. Washout Period (Oral BCAAs)

In this phase, the participants endured a cessation period of 8 weeks, when supplementation of the ME + BCAAs and BCAAs groups was suspended in order to verify whether the supposed benefits of BCAAs were maintained or lost [21].

#### 3.7.3. Exercise Intervention (Phase 1)

The exercise program was divided in two interventions of 16 weeks each, separated by an 8-week detraining (washout) period. Exercise sessions were offered twice a week, with an interval of 36 h for adequate physiological recovery and rest. The exercise protocol respected the guidelines of exercise prescription for older persons and the guidelines of exercise periodization by the American College of Sports Medicine (ACSM) [47,48]. The program started with an adaptation period of 2 weeks, in which seven different exercises were performed using elastic bands (TheraBand^®^, Hygenic Corporation, Akron, OH, USA). The participants were closely supervised for two initial sessions aiming for equipment familiarization and adjustments to the Rating Perceived Exertion (RPE OMNI) scale [49]. During these familiarization sessions, the participants learned the correct technique of the exercises, and selected the proper color, length, and grip width of the elastic bands. The exercise intensity was indirectly calculated using the Karvonen’s formula to predict the target heart rate (HR), with HR_max_ being calculated by an adjusted formula for older persons [50].
HR = ((HR_max_ − resting HR) × %Intensity) + resting HR(1)

After the adaptation period, the exercise program was progressively intensified by increments in both the number of exercises (from 8 to 10 exercises during the rest of the exercise intervention) and the proposed physical effort, imposed by different intensity color bands, according to the OMNI table [49]. The elastic-band exercises applied in the Phase 1 period are shown in Table 1. For safety reasons, the exercise programs were also monitored using heart rate monitors (Polar M200; Polar Electro Oy, Kempele, Finland). Additionally, intensity was measured through the specific rating perceived exertion (RPE) scales for each exercise program [51]. The RPE used is an arbitrary scale ranging from 0 to 10 points, with identical intervals and with reference to the quality of effort: (0) Nothing at all; (1) very weak; (2) weak; (3) moderate; (4) somewhat strong; (5–6) strong; (7–9) very strong; (10) very, very strong (almost maximal).

#### 3.7.4. Washout (ME Detraining)

In this phase, the participants endured a detraining period of 8 weeks, when the ME programs were suspended. The aim was to check if the physiological adaptations acquired during the first phase of ME were maintained or if an 8-week interruption was able to revert the possible effects on immune changes [52].

#### 3.7.5. Exercise Retraining Protocol

The phase 3 (exercise retraining) protocol was also based on the resistant TheraBand (TheraBand^®^, Hygenic Corporation, Akron, OH, USA) elastic bands (Table 2), but included walking, steps, and balance exercises (sometimes with dumbbells and ankle/wrist weights) to compose a multicomponent exercise program for an identical 16-week period (twice a week, on alternate days, also totalizing 32 sessions). The multicomponent program (Table 2) was described by Furtado et al. [53]. The phase 3 program aimed to reproduce most of the daily activities of the older persons in this study [54].

### 3.8. Statistical Analysis

The descriptive statistics for each group, at the baseline and follow-up evaluations, were reported as the mean plus standard deviation (M ± SD), except when mentioned otherwise. All the variables were checked for the normally residual distribution and values were logarithmically transformed when appropriate. One-way Analysis of Variance ANOVA was used to determine baseline differences between the four groups in all the parameters. Effects of time, group, and time x group interactions were assessed through repeated measures ANOVA and Bonferroni post-hoc for multiple comparisons. Additionally, univariate analysis was performed using the paired *t*-test for comparisons during the first phase of interventions (T1 vs. T2). All statistical analyses were performed using the SPSS 21 (SPSS Inc., Chicago, IL, USA), and the level of significance was set at *p* < 0.05.

## 4. Results

The dynamics of the MIP groups and drop-outs are presented in detail in Figure 1. From the 50 (100%) participants initially selected, only 35 participants (70%) completed the intervention. This is an expected experimental loss, as reported by several previous studies [55]. None of the dropouts left the intervention due to injuries or adverse responses. Reported deaths were due to acute events triggered by chronic clinical conditions. Table 3 shows the characterization of participants by MIP groups at the baseline, including nutritional, cognitive, frailty, anthropometric, and body composition status. No statistically significant differences in all the variables appeared, expect for time in residential care and nutritional status assessed by MNA (*p* < 0.05). However, all the groups were within the well-nourished category.

### 4.1. Biochemical Analysis

Table 4 shows the results for IL-10, TNF-α and TNF-α/IL-10 ratio, MPO, albumin, 5TSS-Test, as well as Fried (score) and MMSE. Concerning the IL-10 levels, a classical anti-inflammatory cytokine, not only no effects of time (*p* = 0.690) or time vs. experimental groups were found (CG, BCAAs, ME, and ME + BCAAs), F(degrees of freedom-df:9, 51) = 1.567, *p* = 150), but also Bonferroni post-hoc comparisons did not result in significant variations between time vs. groups (*p* > 0.05). Regarding the TNF-α levels, although we did not observe any interference of time on these pro-inflammatory cytokine levels (*p* > 0.05, Table 4, repeated ANOVA analyses revealed significant interactions between time vs. groups: F(df: 6.758, 47.303) = 2.524, *p* = 0.029. In addition, Bonferroni post-hoc comparisons showed not only higher TNF-α values in the ME + BCAAs group between T2 and T3 (*p* = 0.01), but also a significant decrease of TNF-α was observed between T3 and T4 within the same experimental group (ME + BCAAs, *p* < 0.01). The TNF-α values were unchanged in all other experimental groups. Regarding the TNF-α/IL-10 ratios, no significant variations were observed over time (*p* = 0.703) or within the interactions (time vs. group, *p* = 0.638).

Concerning MPO activity, Table 4 shows that this biomarker was not influenced by time (T1, T2, T3, and T4), except for a slight tendency regarding interactions (time vs. group): F(df: 9, 48) = 2.010, *p* = 0.059. Particularly, the Bonferroni post-hoc comparisons showed that the BCAAs group presented higher MPO activity after re-supplementation (T4) than the values found in the T2 time-point (after the first 16 weeks of the supplementation period, *p* = 0.026). No significant alterations in the MPO activity were observed in other comparisons between groups.

In terms of serum albumin (Table 4), a statistically significant difference in the effect of time was found (F(df: 1949; 46,784) = 3.841, *p* = 0.02), but no other (time vs. group) significant difference was detected between the albumin levels (*p* = 0.219). The pairwise comparison using Bonferroni post-hoc showed a decrease of albumin levels in the BCAAs group in the T3 time-point (after the washout period, *p* = 0.04) as compared to the values found in T1, whereas no other significant variations were observed in the other groups (*p* > 0.05).

### 4.2. Five-Times-Sit-to-Stand-Test (5TSS test)

Table 4 shows no effect of time (*p* = 0.841) or interactions (time vs. group, *p* = 0.846) on the time elapsed to perform the 5TSS test. However, post-hoc adjustments showed that the ME + BCAAs and BCAAs groups presented a significant reduction of the time elapsed to perform this test at time-points T2, T3, and T4 (*p* = 0.009, *p* = 0.014, and *p* = 0.024, respectively).

### 4.3. Cognitive Assessment

The results obtained in the cognitive profile (Table 4), show that, at baseline (T1), 65.7% of the participants (*n* = 23) scored below the 24-point threshold in the MMSE test, indicating that a significant fraction of participants was within the mild/moderate cognitive impairment classification. In addition, at the same time-point (T1), significant differences were found for the cognitive score between the ME + BCAAs group and the other groups (*p* < 0.05). An effect of time (F(df: 3, 93) = 4.262, *p* = 0.007), but not interaction (time vs. group, *p* = 0.296), was observed for the MMSE results. The cognitive MMSE scores increased in the control group between T1 and T2 but decreased subsequently in T3 and T4 (*p* = 0.008). No significant alterations were observed in the other groups. At baseline, 45.7% of the participants were classified as frail and 54.3% as pre-frail.

## 5. Discussion

This study evaluated the effects of exercise and BCAAs on biomarkers of immunity, total albumin, and the cognitive profile of institutionalized older persons. The main findings were that ME showed more proemint result, particullary with BCAA in the improve cognitive profile and muscle strength-related albumin levels in plasma and diminish the frailty status. Moreover, exercise induced slight changes on the pro-inflammatory marker TNF-α.

Albumin levels tend to decrease with age, and this effect seems to imply an increased risk of complications and higher rate of mortality, morbidity, and disabilities such as sarcopenia and frailty [56]. Despite the key participation of albumin on the pH balance and ionic homeostasis in blood, most of the free fatty acid (and some other lipids) transport in the bloodstream is also performed by serum albumin [57]. Not surprisingly, the age-related impaired albuminemia and elevated serum anion gap are known to be associated with hypertension, low cardiorespiratory fitness, and decreased renal function, which are common morbidities of advanced aged people [58]. Therefore, interventions that aim to sustain (or even increase) albuminemia in older persons could represent an important strategy to mitigate the harmful effects of aging and its comorbidities. In this respect, some studies have already shown that BCAAs apparently increases albumin levels in older persons suffering from malnutrition [59].

Our results showed that the serum albumin levels were efficiently sustained or even augmented, in exercising participants (both ME and ME + BCAAs groups) during the first 16 weeks of intervention (phase 1). However, the withdrawal of BCAAs during the washout period (phase 2) quickly decreased those albumin levels, especially in the BCAAs group. The prominent effect of exercise on albumin levels was evident since its levels in both ME and ME + BCAAs groups were fully restored after the phase 3 period (T3 to T4 time-points), whereas only partial recoveries were observed in albumin levels in the BCAAs group at the same time-point. Low serum albumin levels were shown to be the most relevant biomarkers associated with poor physical strength in the older persons [60].

It is broadly accepted that the regular practice of exercise training imposes metabolic, endocrine/physiological, immune, and cognitive adaptations that, among many benefits, can increase skeletal muscle mass and strength, thus, circumventing the deleterious effects of sarcopenia in older persons [61].

The chronic exercise-mediated adjustments on insulin/glucagon balance, thyroid, and steroid hormones, such as testosterone, cortisol, and estrogens, can also be involved in the enhancement of hepatic and protein muscle metabolism (proteolysis, proteogenesis, and protein turnover), with clear consequences on the circulating amino acid levels (e.g., glutamine and alanine), blood pH and electrolyte balance (hydric/ionic homeostasis), and renal functions [62].

However, it was reported that the putative effect of amino acid/protein supplementation in older women could be masked by sufficient daily protein intake, as we attested in all institutionalized participants in this study [63]. Thus, the proper mechanism behind this effect still needs to be fully understood for this special population. In fact, to our knowledge, this is the first study to show the potential of physical exercise associated or not with BCAAs supplementation to maintain serum albumin levels in older persons living in RCH.

Contrarily to the albumin results, the monitored inflammatory markers (IL-10, TNF-α, and MPO) did not show significant alterations over time. Apparently, we can putatively suggest, that the physical exercise intensities reached in the sessions, as well as the BCAAs supplementation effect compared to the daily protein intake in this population, were not sufficient to induce a significant impact on the inflammatory status in the participants in this study. Other interventions with older persons have been able to show a strong anti-inflammatory effect of exercise training, but it seems that these results were observed for intervention periods longer than 16 weeks [43,44].

Interestingly, even though an increase in the levels of the pro-inflammatory cytokine TNF-α was observed in the ME + BCAAs group from T1 to T2 and T3, this finding was accompanied by a proportional increase of the anti-inflammatory cytokine IL-10, since the TNF-α/IL-10 ratio was not different in this group over time. Moreover, at the end of the intervention, TNF-α levels significantly decreased in this group. In accordance with the literature, IL-10 is a key anti-inflammatory cytokine that acts by inhibiting systemic inflammation mediated by TNF-α [64].

Concomitantly, BCAAs alone did not induce alterations in both IL-10 and TNF-α levels. These results differ slightly from what is observed in the literature regarding this type of intervention on inflammatory status [65]. Based on the literature, there is a close interaction between the inflammatory status and aging, and in this respect, it is widely accepted that older persons, especially sedentary people, present a chronic, systemic, sterile low-grade inflammation associated with aging, a phenomenon named inflammaging [66]. It is highlighted that inflammaging plays an important role in the loss of lean mass, which leads to sarcopenia and frailty, as well as increases the risk of the development of diseases and comorbidities, such as cognitive decline, atherosclerosis, insulin resistance, etc. [67].

Despite the fact that literature defines the ability to induce an anti-inflammatory change as a hallmark of physical exercise, in general, our results did not corroborate this fact. It is paramount to mention that some factors could putatively influence the lack of significant results in the inflammatory analysis. Firstly, the occurrence of inflammaging and pathophysiological disturbances in our participants could be crucial for the response magnitude observed during the interventions here. Second, the low level of physical activity of our participants before the interventions could mitigate the benefits that would be achieved with the physical exercise sessions and, consequently, limit physiological adaptation. These factors, associated with polypharmacy, a high rate of comorbidities, and the small sample size that finished the study, may determine the lack of significant effects observed.

There is a consensus in the literature that physical exercise sessions stimulate the release of cytokines, such as IL-6, IL-10, and TNF-α, in response to contracting skeletal muscles, which are responsible not only for tissue restoration and energy metabolism, but also for the adjustment of the systemic inflammatory status [68].As appealing as these effects are, physical exercise training also improves human antioxidant defenses as observed in several studies which may also justify the use of exercise interventions to counteract the progression of oxidative-related diseases [69].

There are solid pieces of evidence that the loss of muscle strength and power in the lower limbs, which is characterized by a decline of up to 50% in overall muscle strength from the age of 30 to 80 years [52,53] is associated with an increased incidence of falls.

Particularly, physical exercise training improves body composition, muscle strength, metabolic parameters, bone health, and functionality as well as reduces the risk of mortality, chronic diseases, cognitive deterioration, falls, and depression [70]. Here, we observed that only the ME + BCAAs group presented an improved physical performance in the 5TSS test. Neither ME or BCAAs alone were sufficient to mediate improvements in lower body strength. Only the combination of exercise and supplementation did so. This result was achieved probably due to multiple factors, from physiological to cognitive positive effects that were not directly assessed by the applied methodology here. According to the literature, the 5TSS test is an important performance test that invokes physical skills and abilities that could have been particularly developed during phase 3 of this study. The phase 3 of our study included walking activities, steps, and balance exercises, which mimic the participants’ regular daily life activities.

It is important to point out that strength exercise training has been proposed as one of the most effective methodologies, presenting best results in bringing back safety in per-forming the common tasks of daily life, focusing on the optimization of neuromuscular function for better benefits [71].

Multicomponent programs combine aerobic and strength exercises, including other physical skills, such as balance and flexibility [54], in order to optimize the functional capacity of frail older persons [72], as well as to maintain their independence to perform basic activities of daily living [73]. Concerning supplementation, it was reported that branched-chain amino acids, particularly L-leucine, showed significant results in inducing hypertrophy in older persons and improving their functional capacity [58,59].

Taking into account that cognitive impairment is one of the main factors that cause morbidity and high health costs worldwide [74], our results show that physical exercise training, in association or not with BCAAs, was able to maintain the cognitive scores of the participants and could have important practical applications. Considering the population enrolled here (pre-frail and frail octogenarians) and the trend for the natural decline of their cognitive functions, the maintenance of those cognitive scores by exercise is, per se, a remarkable achievement. The literature supports the positive effect of BCAAs in older persons, to improve their mood state [75], the perception of fatigue, and their performance in a mental task [76], which are abilities that were not evaluated here. Leucine is important since it activates the mammalian target of rapamycin complex 1 (mTORC1) and the downstream phosphorylation of p70S6 kinase and 4E (eIF4E)-binding protein 1 (4E-BP1) and related signaling pathways [77]. The aging muscle is less responsive to lower doses of amino acids when compared to the young muscle and may require higher quantities of protein to acutely stimulate equivalent muscle protein synthesis [78]. Nevertheless, the dose and duration of BCAAs proposed here did not affect the cognition scores in our participants.

### Study Limitation and Perspectives for Future Researchers

The entire study was conducted with human octogenarians and, given the difficulty to control several influencing factors in this type of population, this study had the additional merit of causing a minimal impact on their daily routines at the residential care homes. In addition, our results here represent real-world data reflecting the reality at residential care homes. We screened participants with disabilities and comorbidities that, although expecting high rate of dropouts and low motivational issues, we could accomplish the proposed goals with a reasonable number of participants. The execution of a controlled study over 40 weeks with such a particular population also introduces other limitations. We suggest that the use of other methods of exercise training, such as the use of playful activities (dance and music sessions) might elevate the adherence of this population to the program.

## 6. Conclusions

This study showed that multicompetent exercise training, with minor effect of BCAAs, triggered alterations in the inflammatory status and physical profiles of older persons, while helping maintain cognitive levels. Taken together, the achieved results, could help increase autonomy and efficiency in the performance of daily activities. Unlike other studies, our results showed that supplementation with BCAA did not induce substantial changes in health-related parameters at older ages. It is possible that the heterogeneity and limited sample size might have limited the statistical relevance of our results. Despite a slight and transient variation over time observed in some inflammatory and cognitive parameters, it is possible that the results here were influenced by the comorbidity status of each group.

## Figures and Tables

**Figure 1 nutrients-13-01106-f001:**
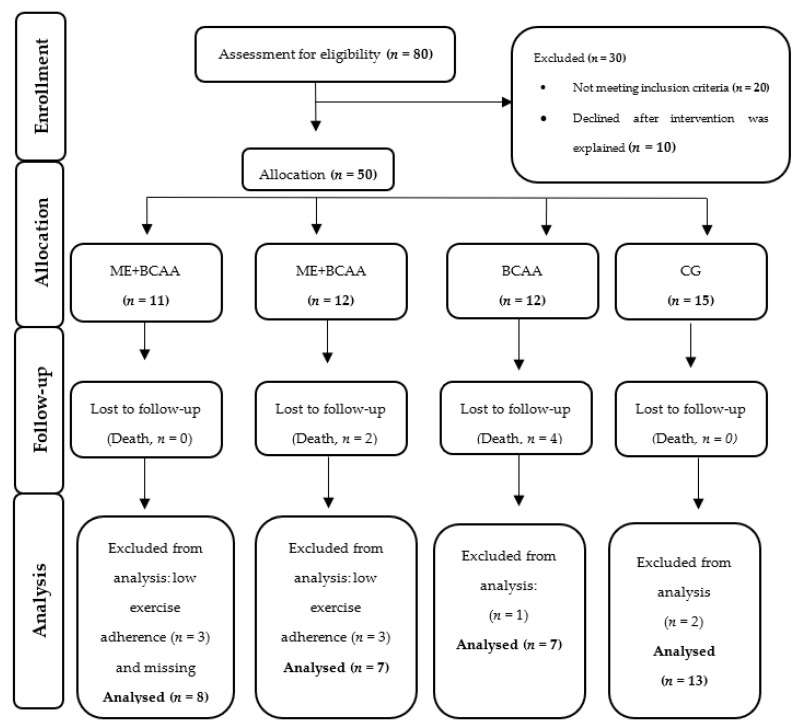
CONSORT Flowchart of study participants [30]. ME + BCAA, Multicomponent Exercise + Branched Chain Amino Acid; ME, Multicomponent Exercise; BCAA, Branched Chain Amino Acid; CG, Control Group.

**Figure 2 nutrients-13-01106-f002:**
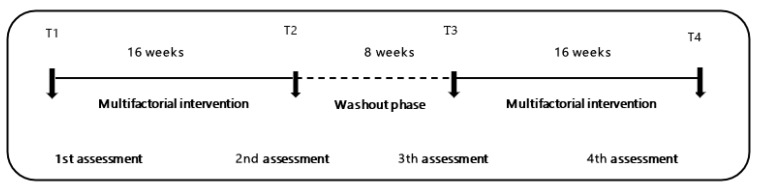
Chronological order of multifactorial interventions study design. T1 to T2 (elastic-band exercise, 16 weeks, 8 weeks), T2 to T3 (wash-out), T3 to T4 (multicomponent exercise, 16 weeks).

**Table 1 nutrients-13-01106-t001:** Example of elastic-band exercise sessions applied in phase 1.

Warm-Up				5 min	PSE 1–3	Progression	Weeks	Intensity (Color)
**Exercises (8–10)**	**Sets**	**Repetitions**	**Cadence**	**Interval**	**PSE**	2 × 10	2	Yellow
Front squat	2–3	10–20	2:3	30–45 s	4 to 6	3 × 20	2	Yellow
Chair unilateral hip flexion	2–3	10–20	2:3	30–45 s	4 to 6	3 × 10	2	Red
Chair Bench over row (with flexion)	2–3	10–20	2:3	30–45 s	4 to 6	3 × 20	2	Red
Chest Press (stand and/or chair)	2–3	10–20	2:3	30–45 s	4 to 6	3 × 10	2	Green
Standing (or chair) reverse fly	2–3	10–20	2:3	30–45 s	4 to 6	3 × 20	2	Green
Shoulder Press/twist arm position	2–3	10–20	2:3	30–45 s	4 to 6	3 × 15	2	Blue
Chair (or stand) frontal total raiser	2–3	10–20	2:3	30–45 s	4 to 6	3–4 × 10^−15^	2	Blue
Biceps arm curl (stand and/or chair)	2–3	10–20	2:3	30–45 s	4 to 6			
Chair Overhead triceps extension	2–3	10–20	2:3	30–45 s	4 to 6			
**Cooling down**				5 min	PSE 1–2			

Notes: PSE—Perception subjective effort.

**Table 2 nutrients-13-01106-t002:** Example of multicomponent exercise sessions applied in phase 2.

Exercises (8–10)	Sets	Repetitions	Cadence	Interval	PSE
Front squat	2–3	10–20	2:3	30–45 s	4 to 6
Chair unilateral hip flexion	2–3	10–20	2:3	30–45 s	4 to 6
Chair Bench over row (with flexion)	2–3	10–20	2:3	30–45 s	4 to 6
Chest Press (stand and/or chair)	2–3	10–20	2:3	30–45 s	4 to 6
Standing (or chair) reverse fly	2–3	10–20	2:3	30–45 s	4 to 6
Shoulder Press/twist arm front position	2–3	10–20	2:3	30–45 s	4 to 6
Chair (or stand) frontal total raiser	2–3	10–20	2:3	30–45 s	4 to 6
Biceps arm curl (stand and/or chair)	2–3	10–20	2:3	30–45 s	4 to 6
Chair Overhead triceps extension	2–3	10–20	2:3	30–45 s	4 to 6
Circuit Training					
Walking around the room	2–3	3 min		30–45 s	4 to 6
Balance/agility exercise	2–3	3 min		30–45 s	4 to 6

Notes: PSE—Perception subjective effort.

**Table 3 nutrients-13-01106-t003:** Characterization of participants by intervention groups at baseline.

Variables	ME + BCAA (*n* = 8)	ME (*n* = 7)	BCAA (*n* = 7)	CG (*n* = 13)	*p*-Value
	M ± SD	M ± SD	M ± SD	M ± SD	
Age (years)	80 ± 6.1	86.7 ± 4	84.2 ± 5.8	83.1 ± 5.4	0.139
Time in residential care (years)	3.6 ± 1	4.7 ± 1.4	4.5 ± 1.1	5 ± 1	0.06
MNA (0–30 pts)	25.5 ± 2.2	24 ± 2.7	21.7 ± 2.8	24.7 ± 1.8	0.02
BMI (kg/m^2^)	28.53 ± 5.1	28.7 ± 5.6	25.8 ± 3.1	30.2 ± 3.7	0.23
Stature (cm)	158 ± 0.05	150 ± 0.06	161 ± 0.12	155 ± 011	0.16
Comorbidity index (0–10 pts)	4.87 ± 1.12	5.28 ± 0.95	5.42 ± 1.1	4.92 ± 1.2	0.71
Schooling time (years)	4 ± 0	4 ± 0	4 ± 0	4 ± 0	0.99
Cognitive profile (0–30 pts)	26.00 (3.11)	21.00 (3.78)	20.85 (2.79)	21.69 (2.89)	0.00
Physical Frailty index (0–5 pts)	2.00 (0.53)	2.71 (1.1)	3.00 (0.57)	2.16 (0.71)	0.40
Daily Individual Protein (gr/kg/day)	1.42 ± 0.28	1.83 ± 0.44	1.48 ± 0.22	1.60 ± 0.23	0.159
BCAAs (per person/gr/week)	30.3 ± 6.0	n.d.	28.4 ± 5.0	n.d.	

Notes: BMI: Body mass index; MNA: Mini nutritional assessment; M ± SD: Mean (standard and deviation); pts: Points; Kg/m^2^: Kilograms; cm: Centimeters; One-way ANOVA was used to compare groups, except for the Comorbidity index (Fisher Exact Test). BCAA Branched Chain Amino Acids.

**Table 4 nutrients-13-01106-t004:** Statistical analysis comparison of four time-points moments of multifactorial intervention for biochemical, cognitive profile, physical frailty index, and functional fitness test.

		Time-Points of Evaluation				Effect	F	Overall *p*
Biomarker/Variables	Groups	T1	T2	T3	T4			
		M ± SD	M ± SD	M ± SD	M ± SD			
**IL-10** (μg/mL)	ME + BCAA	10.36 (6.96)	12.0 (6.53)	15.99 (7.98)	11.52 (7.56)			
	ME	8.68 (7.68)	12.25 (12.35)	4.16 (3.39)	10.53 (5.82)	Time	0.491	0.690
	BCAA	7.71 (2.54)	9.24 (4.15)	13.83 (6.94)	9.85 (10.89)	Time*group	1.567	0.150
	CG	16.10 (7.4)	12.21 (2.81)	12.74 (7.36)	20.45 (5.42)			
**TNF-α** (pg/mL)	ME + BCAA	62.44 (53.65)	71.42 (38.06)	112.86 (62.51)	57.37 (31.18)			
	ME	41.78 (54.08)	45.83 (21.07)	24.92 (15.60)	54.05 (29.19)	Time	1.552	0.210
	BCAA	32.65 (15.74)	37.18 (26.91)	62.93 (35.77)	60.02 (55.42)	Time*group	2.524	0.015
	CG	44.46 (41.72)	44.81 (37.16)	41.78 (37.86)	57.01 (44.15)			
**TNF-α/IL-10 ratio** (pg/mL)	ME + BCAA	6.24 (4.46)	7.47 (4.09)	6.96 (1.63)	6.10 (3.25)			
	ME	4.43 (1.99)	9.06 (10.46)	8.64 (7.36)	5.70 (3.27)	Time	0.472	0.703
	BCAA	5.44 (3.39)	3.85 (1.84)	5.45 (1.54)	11.19 (9.77)	Time*group	0.777	0.638
	CG	4.10 (1.27)	5.37 (1.56)	4.56 (1.80)	4.41 (0.38)			
**MPO** (μg/mL)	ME + BCAA	5653.91 (1106.71)	5871.97 (1159.09)	4843.50 (1221.63)	5196.53 (591.62)			
	ME	5935.71 (1315.33)	5252.76 (1084.06)	4685.42 (1043.31)	4512.34 (794.61)	Time	1.191	0.323
	BCAA	5139.04 (909.07)	4069.64 (1009.10)	5416.47 (1539.50)	5575.80 (1181.43)	Time*group	2.010	0.059
	CG	4623.56 (699.03)	4593.56 (1310.34)	4655.42 (815.10)	4327.39 (863.95)			
**Albumin** (g/dL)	ME + BCAA	3.60 (0.39)	3.63 (0.61)	3.82 (0.54)	3.75 (0.63)			
	ME	3.73 (0.61)	4.12 (0.74)	3.57 (0.43)	4.13 (0.22)	Time	3.841	0.013
	BCAA	3.77 (0.39)	3.61 (0.40)	1.56 (2.15)	2.83 (1.60)	Time*group	1.446	0.185
	CG	3.75 (0.72)	3.60 (0.35)	2.59 (1.85)	2.96 (1.69)			
**5TSS test** (s)	ME + BCAA	21.87 (3.64)	18.71 (3.59)	20.66 (4.98)	17.54 (4.4)			
	ME	26.69 (12.98)	28.02 (11.28)	26.08 (10.46)	27.56 (12.24)	Time	0.165	0.841
	BCAA	36.54 (14.14)	36.24 (13.39)	36.74 (11.89)	35.76 (17.28)	Time*group	0.436	0.846
	CG	24.58 (8.99)	24.76 (9.0)	23.66 (9.30)	25.17 (9.75)			
**Physical Frailty** (index)	ME + BCAA	2.00 (0.53)	1.50 (0.53)	2.12 (0.99)	2.00 (0.53)			
	ME	2.71 (1.1)	2.57 (1.13)	2.14 (0.69)	2.00 (0.81)	Time	2.702	0.05
	BCAA	3.00 (0.57)	2.14 (0.37)	2.28 (1.25)	2.71 (0.48)	Time*group	**3.799**	**0.00**
	CG	2.16 (0.71)	2.25 (0.75)	2.66 (0.49)	3.16 (0.71)			
**MMSE** (0–30 points)	ME + BCAA	26.00 (3.11)	26.37 (2.44)	26.00 (2.87)	24.37 (3.58)			
	ME	21.00 (3.78)	22.42 (2.99)	21.00 (4.65)	20.00 (3.91)	Time	**4.262**	**0.13**
	BCAA	20.85 (2.79)	19.42 (4.07)	20.71 (4.02)	19.57 (3.64)	Time*group	1.214	0.305
	CG	21.69 (2.89)	23.92 (3.47)	23.23 (3.83)	21.76 (2.94)			

Notes: M ± SD: Mean (standard and deviation); ME: Multicomponent exercise; BCAA: Branched-chain amino acids; IL: Interleukin; TNF-α: Tumor Necrosis Factor-alpha; MPO: Myeloperoxidase; MMSE: Mini Mental State Exam; 5TSS test: Five-Times-Sit-to-Stand-Test; T1 to T2 (elastic-band exercise, 16 weeks, 8 weeks), T2 to T3 (wash-out), T3 to T4 (multicomponent exercise, 16 weeks). * time versus group interactions. Statistically significant differences are denoted in bold.

## Data Availability

The data presented in this study are available on request from the corresponding author. Data supporting the reported results is the property of CIDAF, Faculty of Sport Sciences and Physical Education, University of Coimbra, Coimbra, Portugal.

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
