# Peer review of "Effect of Training-Detraining Phases of Multicomponent Exercises and BCAA Supplementation on Inflammatory Markers and Albumin Levels in Frail Older Persons"

_nutrients, 2021, doi:10.3390/nu13041106_

Round 1

Reviewer 1 Report

It is a very interesting approach. A number of independent parameters have been included in this study that play an important part in the inflammatory status and age- related conditions.

I feel that a more specific reference to the subject of sarcopenia would be useful.  I would suggest that the authors would include some additional literature.

A minor check of the use of the language is required.

Author Response

Dear reviewer #01,

I send below all the questions answered point-to-point, following recommendation from the Nutrients editorial office:

R#1: English Language and style (English language and style are fine/minor spell check required)

Authors: Thank you for your response. After reviewing the article by all co-authors following the indications of the 3 reviewers, the language was duly verified. 

R#2: Does the introduction provide sufficient background and include all relevant references? (can be improved)

Authors: Thank for your suggestion. We made a substantial increase in the introduction section, strengthening some concepts (making them more precise) and even mentioning some additional studies.

R#3: It is a very interesting approach. A number of independent parameters have been included in this study that play an important part in the inflammatory status and age- related conditions.

Authors:  Thank you for your observation. 

R#5: A minor check of the use of the language is required.

Authors: As mentioned above, the language has been duly improved 

Reviewer 2 Report

Dear authors, 

This is a very interesting and enjoyable to read paper on the important topic of maintaining muscle mass and functional ability of older people. I agree that this is an important topic of investigation. I have some minor suggestions to improve the manuscript. 

Line 25: “Older people”. Suggest splitting this sentence into two.

Line 29: “associated or not” – suggest revise this sentence for clarity.

Line 37: no full stop.

There is not enough focus on the actual statistical results in the abstract.

Line 49: Should be “Physical exercise”.

Line 52: Please avoid use of “elderly”. Many people aged over 65 continue to work nowadays and do not like to be referred to as this. Older people or older persons are fine to continue using. Relevant for whole manuscript.

Line 54: Now individuals is used. Subjects is also used later and then older adults. Please remain consistent as per suggested above.

Line 73: If you are going to abbreviate residential care homes, please do from the first instance in line 68.

Line 75: Please do not refer to as institutions as this is also outdated and no longer appropriate. Residential care homes is fine.

Line 80: Time points is more appropriate than moments.

Line 82: Should be “totalling”.

Line 108: Were all meals provided? Was it confirmed that all meals were consumed? How was this measured?

Line 119: Was there any flavouring as the taste of BCAA’s can be quite off-putting?

Line 158: I believe these devices contain data on HRV. This could be an interesting follow-up study.

Line 193: Why say weakness when you can say strength? Hand-grip strength specifically.

Line 216: Suggest reworded the next few lines to appropriate tense.

Line 254: Please provide all inter and intra assay variations here rather than in the results. A brief justification of the selection of these biomarkers should be provided in the introduction.

Line 319: There is over use of “In relation to…”.

Line 329: 45.7% instead of 45,7%.

Line 330: The last sentence is incomplete.

Line 411: Polypharmacy

Line 444: Given previous literature on the anabolic nature of leucine and it’s previous use in older people and in potential to prevent or reduce anabolic resistance, I feel this should be briefly mentioned earlier – perhaps even in the discussion to introduce the importance of BCAA’s/leucine as a potential way to prevent muscle wasting and sarcopenia.

Line 450: frail

Line 458: Limitations should come before the conclusions.

Line 465: Statistical power. Was a power analysis completed before the study started?

Line 473: Delete “a”.

While the focus is on BCAA, could other nutrients be included? For example some research has suggested benefits in similar studies that use vitamin D and omega-3 fatty acids. It would be useful to at least briefly mention this line of thinking in the discussion and to cite appropriately. See Stuart Phillips.

Author Response

Dear reviewer #02,

Below are all the questions answered point-to-point, following recommendation from the Nutrients editorial office.

This is a very interesting and enjoyable to read paper on the important topic of maintaining muscle mass and functional ability of older people. I agree that this is an important topic of investigation. I have some minor suggestions to improve the manuscript. 

Author Response: Thank you for appreciating our research.

R#1: Line 25. “Older people”. Suggest splitting this sentence into two.

Authors: Thank you for your observation. We split this sentence in accordance with your suggestion.

R#2: Line 29: “associated or not” – suggest revise this sentence for clarity.

Authors: Thank you for your observation. We have revised the sentence.

R#3. Line 37: no full stop.

Authors: Thank you for your observation. We made the corrections.

R#4: There is not enough focus on the actual statistical results in the abstract.

Authors: Thank you for your suggestion We altered the abstract.

R#5: Line 49: Should be “Physical exercise”.

Authors: Thank you for your observation. We made the alteration.

R#6: Line 52: Please avoid use of “elderly”. Many people aged over 65 continue to work nowadays and do not like to be referred to as this. Older people or older persons are fine to continue using. Relevant for whole manuscript.

Authors: Thank you for your suggestion. In fact, it seems that “older people” or “older persons” are better. We proceeded to corrections and replaced this expression in the entire text.

R#7: Line 54: Now individuals is used. Subjects is also used later and then older adults. Please remain consistent as per suggested above.

Authors: Thank you for your observation. We made the alterations as suggested.

R#8: Line 73: If you are going to abbreviate residential care homes, please do from the first instance in line 68.

Authors: Thank you for your attention. We corrected the first appearance of this abbreviation and adjusted in the whole text accordingly.

R#9: Line 75: Please do not refer to as institutions as this is also outdated and no longer appropriate. Residential care homes is fine.

Authors: Thank you for your observation. We made the alterations as suggested.

R#10: Line 80: Time points is more appropriate than moments.

Authors: Thank you for your suggestion. We did the suggested changes.

R#11: Line 82: Should be “totalling”.

Authors: Thank you for your observation, we made the correction.  

R#12: Line 108: Were all meals provided? Was it confirmed that all meals were consumed? How was this measured?

Authors: Thank you for your concern. The daily meals are controlled by the nutritional staff of the RCH. In fact, it is not easy to control the food intake of all participants, however, since we were constantly in contact with the registered nutritionist and co-workers of the RCH, we also check for any changes/alterations in the participant’s appetite. If the staff saw that there was food left over, and that consumption dropped from what was usually being consumed, they were asked to let us know about it. However, no major changes were reported during the duration of the study.

R#13: Line 119: Was there any flavoring as the taste of BCAA’s can be quite off-putting?

Authors: Thanks for asking. The BCAA supplement provided did not contained any flavorings or smell. We also now added a sentence in the methods to mention that. However, we did not see any drop-outs due to this.

R#14: Line 158: I believe these devices contain data on HRV. This could be an interesting follow-up study.

Authors: Thank you for the suggestion. Definitely some to pursue in the future.

R#15: Line 193: Why say weakness when you can say strength? Hand-grip strength specifically.

Authors: Thank you for your observation. We made the adjustments in the manuscript.  

R#16: Line 216: Suggest reworded the next few lines to appropriate tense.

Authors: Thank you for your observation. We made the corrections in the manuscript. 

R#17: Line 254: Please provide all inter and intra assay variations here rather than in the results. A brief justification of the selection of these biomarkers should be provided in the introduction.

Authors: Thank you for your attention. We re-organized the place to mention the intra assay variation in accordance with your suggestion. In fact, that is better located now. We also justify the use of those biomarkers in the introduction.

R#18: Line 319: There is over use of “In relation to…”.

Authors: Thank you for your observation, we made the corrections. 

R#19: Line 329: 45.7% instead of 45,7%.

Authors: Thank you for your observation, we made the alteration.

R#20: Line 330: The last sentence is incomplete.

Authors: Thank you for your observation. We have reworded it in the manuscript.

R#21: Line 411: Polypharmacy

Authors: Thank you for your suggestion. We made the alteration.

R#22: Line 444: Given previous literature on the anabolic nature of leucine and it’s previous use in older people and in potential to prevent or reduce anabolic resistance, I feel this should be briefly mentioned earlier – perhaps even in the discussion to introduce the importance of BCAA’s/leucine as a potential way to prevent muscle wasting and sarcopenia.

Authors: Thank you for your observation. We have adjusted the text to reflect this suggestion.

R#23: Line 450: frail

Authors: Thank you for your observation, we altered it.

R#24: Line 458: Limitations should come before the conclusions.

Authors: Thank you for your observation, we altered it.

R#25: Line 465: Statistical power. Was a power analysis completed before the study started?

Authors:  Thank you for your observations. In fact, we calculated power analysis a priori using G*Power. We established alpha (type I error rate) as 0.5, and power (type II error rate) as 0.80, giving a total of 24 participants as sufficient to acquire sufficient power. We recruited more participants (as can be inferred in the Figure 1 – Flowchart), and also considering a normal dropping-out rate to achieve what was supposed in the beginning of the study.

R#26: Line 473: Delete “a”.

Authors: Thank you for your observation, we altered it.

R#27: While the focus is on BCAA, could other nutrients be included? For example some research has suggested benefits in similar studies that use vitamin D and omega-3 fatty acids. It would be useful to at least briefly mention this line of thinking in the discussion and to cite appropriately. See Stuart Phillips.

Authors: : Thank you for your observation, we have approached it in the discussion.

Reviewer 3 Report

Comments to the Authors of manuscript number: nutrients-1147227 entitled “Effect of training-detraining phases of multicomponent exercises and BCAA supplementation on inflammatory markers and albumin levels in older persons”.

Authors presented the study very well organized and described, however it should be corrected. I recommend minor revision.

Abstract

• The abstract succinctly provides the overview, general findings, and the hypotheses generated from the study.

Introduction

• The introduction effectively places the study and its importance in relation to the identified gaps in the scientific understanding of BCAA supplementation with training-detraining phases of multicomponent exercises in older people. • at the present time, society is getting older. Aging is an inherent process in human life, but it can progress at different rates. Poor physical activity in the elderly due to comorbidities or primary sarcopenia increases the risk of falls and fractures. especially when osteoporosis is diagnosed. Immobilization in this case can lead to higher mortality. Thus, in general the economic cost of care increases. • The goal of the study is clearly identified.

Methods

The attached graphics help someone understand the complicated study design. At the beginning there were 80 subjects. Finally, only 35 people were involved. There is suggestion to add in what season this study was performed. It is know that seasons can influence the physical activity and mood of subjects. Eligibility criteria were properly defined.

Moreover, exercise protocol is presented in details. Three phases are included, two are presented in tables. Authors wrote that International PA Questionnaire short version (IPAQ-SV) was used. It is worth to indicate by whom it was authorized. It is the same in relation to 5TSS-Test and Mini Nutritional Assessment (especially that this last uses strictly described points). Authors used Mini Mental State Examination, adapted for the Portuguese language. By whom it was authorized?

Results and discussion

• The results are very well presented, then discussed. The results section concisely pointed out important findings which were detailed in the Tables. The description and interpretation of the obtained results do not raise any question.

I have one question: there is no information about injures. was study ruined by an injury in some point?

Author Response

Dear reviewer #03,

I send below all the questions answered point-to-point, following recommendation from the Nutrients editorial office:

Abstract

 The abstract succinctly provides the overview, general findings, and the hypotheses generated from the study.

Authors: Thank you for you commnets. However, we made substantial changes to the summary, introducing essential information for understanding our study.

Introduction

The introduction effectively places the study and its importance in relation to the identified gaps in the scientific understanding of BCAA supplementation with training-detraining phases of multicomponent exercises in older people. • at the present time, society is getting older. Aging is an inherent process in human life, but it can progress at different rates. Poor physical activity in the elderly due to comorbidities or primary sarcopenia increases the risk of falls and fractures. especially when osteoporosis is diagnosed. Immobilization in this case can lead to higher mortality. Thus, in general the economic cost of care increases. • The goal of the study is clearly identified.

Authors: Thank you for you commnets. We made substantial changes to the introduction section following  our suggestions , and introducing more information to reinforce the state of the art

Methods

R#1: The attached graphics help someone understand the complicated study design. At the beginning there were 80 subjects. Finally, only 35 people were involved. There is suggestion to add in what season this study was performed. It is know that seasons can influence the physical activity and mood of subjects. Eligibility criteria were properly defined.

Authors: The study began in the spring. We have added that information to the manuscript.

R#2: Moreover, exercise protocol is presented in details. Three phases are included, two are presented in tables. Authors wrote that International PA Questionnaire short version (IPAQ-SV) was used. It is worth to indicate by whom it was authorized. It is the same in relation to 5TSS-Test and Mini Nutritional Assessment (especially that this last uses strictly described points). Authors used Mini Mental State Examination, adapted for the Portuguese language. By whom it was authorized?

Authors: Yeah, for use all the questionnnaries we previouslly received authorization by the authors. 

Results and discussion

R#3: The results are very well presented, then discussed. The results section concisely pointed out important findings which were detailed in the Tables. The description and interpretation of the obtained results do not raise any question.

Authors: Thank you for you commnets

R#4. I have one question: there is no information about injures. was study ruined by an injury in some point?

Authors: No injuries occurred during the intervention period due to the exercise program application, and this fact was well described in the results section.
